# Potential Neuroprotective Effects of Dietary Omega-3 Fatty Acids on Stress in Alzheimer’s Disease

**DOI:** 10.3390/biom13071096

**Published:** 2023-07-08

**Authors:** Kaitlyn B. Hartnett, Bradley J. Ferguson, Patrick M. Hecht, Luke E. Schuster, Joel I. Shenker, David R. Mehr, Kevin L. Fritsche, Martha A. Belury, Douglas W. Scharre, Adam J. Horwitz, Briana M. Kille, Briann E. Sutton, Paul E. Tatum, C. Michael Greenlief, David Q. Beversdorf

**Affiliations:** 1School of Medicine, University of Missouri-Columbia, Columbia, MO 65212, USA; hartnettk@health.missouri.edu; 2Interdisciplinary Neuroscience Program, University of Missouri-Columbia, Columbia, MO 65212, USA; fergusonbj@health.missouri.edu (B.J.F.); patrick.m.hecht@gmail.com (P.M.H.); 3Department of Health Psychology, University of Missouri-Columbia, Columbia, MO 65212, USA; 4Department of Neurology, University of Missouri-Columbia, Columbia, MO 65212, USA; shenkerj@health.missouri.edu; 5School of Medicine, University of Kansas, Kansas City, KS 66160, USA; lschuster@kumc.edu; 6Family & Community Medicine, University of Missouri-Columbia, Columbia, MO 65212, USA; mehrd@health.missouri.edu; 7Department of Nutrition and Exercise Physiology, University of Missouri, Columbia, MO 65211, USA; klfritsche37@gmail.com; 8Department of Human Sciences, Ohio State University, Columbus, OH 43210, USA; belury.1@osu.edu; 9Department of Neurology, Ohio State University, Columbus, OH 43210, USA; doug.scharre@osumc.edu; 10Strock Medical Group, Boulder, CO 80303, USA; adamjonathanhorwitz@gmail.com; 11Children’s Hospital Colorado, Aurora, CO 80045, USA; bkille01@gmail.com; 12College of Osteopathic Medicine, William Carey University, Hattiesburg, MS 39401, USA; briann7sutton@aol.com; 13Division of Palliative Medicine; Washington University. St. Louis, MO 63110, USA; paultatum@wustl.edu; 14Department of Chemistry, University of Missouri, Columbia, MO 65211, USA; greenliefm@missouri.edu; 15Department of Radiology, University of Missouri, Columbia, MO 65212, USA; 16Psychological Sciences, University of Missouri, Columbia, MO 65212, USA

**Keywords:** Alzheimer’s disease, stress, omega-3 fatty acids

## Abstract

Background: A large number of individual potentially modifiable factors are associated with risk for Alzheimer’s disease (AD). However, less is known about the interactions between the individual factors. Methods: In order to begin to examine the relationship between a pair of factors, we performed a pilot study, surveying patients with AD and controls for stress exposure and dietary omega-3 fatty acid intake to explore their relationship for risk of AD. Results: For individuals with the greatest stress exposure, omega-3 fatty acid intake was significantly greater in healthy controls than in AD patients. There was no difference among those with low stress exposure. Conclusions: These initial results begin to suggest that omega-3 fatty acids may mitigate AD risk in the setting of greater stress exposure. This will need to be examined with larger populations and other pairs of risk factors to better understand these important relationships. Examining how individual risk factors interact will ultimately be important for learning how to optimally decrease the risk of AD.

## 1. Introduction

Alzheimer’s disease (AD) is the most common form of dementia, having a major impact on society, and in particular, a significant impact on caregivers. Limited treatment options are available, so targeting potential modifiable risk factors to prevent the development of AD is of significant interest. While genetic risk factors for AD are well known, a variety of potentially modifiable risk factors have also been demonstrated. Furthermore, due to the major impact on patients and their families the resulting loss of quality of life for the patient, it is critical to identify how individual modifiable factors might interact in order to optimize preventative approaches. Current approved therapies, such as the cholinesterase inhibitors and NMDA antagonists, offer limited help [1]. There are newer agents that target pathogenesis of diseases such as AD, but cost and uncertainties regarding their clinical effects currently limit their impact [1]. Therefore, a better understanding of modifiable risk factors is essential.

The Lancet Commission identified a 12-factor life-course model for modifiable factors that may affect dementia outcomes [2]. In the Lancet study, less education, hypertension, hearing impairment, smoking, obesity, depression, physical inactivity, diabetes, infrequent social contact, excessive alcohol consumption, head injury, and air pollution exposure collectively contributed to about 40% of the risk for dementia [2]. Additionally, an extensive review of “successful cognitive aging” strategies supported regular physical activity, treatment of cardiovascular risk factors (hypertension, diabetes, hyperlipidemia, obesity, metabolic syndrome), cognitively stimulating activities, social engagement, a heart-healthy diet, smoking cessation, managing stress and depression, getting adequate sleep, avoiding anticholinergics, assessing sensory deficits, limiting alcohol use, and avoiding physical and toxin-related brain damage as protective [3]. In a study tracking patients with type 2 diabetes, greater adherence to healthy lifestyle factors (no current smoking, moderate alcohol consumption, regular physical activity, healthy diet, adequate sleep, less sedentary behavior, and frequent social contact) was also found to be associated with decreased incidence of dementia [4]. When dementia is already in place, however, there is a more limited set of factors that impact the course [5].

One of the most consistent findings impacting the risk of Alzheimer’s disease dementia or neurocognitive decline more generally is the impact of a “Mediterranean diet” [6,7,8,9] high in intake of vegetables, legumes, fruits, nuts, cereals, and unsaturated fatty acids (mostly olive oil), with moderate-to-high fish intake, low-to-moderate intake of dairy, low intake of meat and saturated fatty acids, and regular but limited alcohol use [9]. The Mediterranean diet is associated with less accumulation of cerebral beta-amyloid in the Australian Imaging, Biomarkers, and Lifestyle Study of Ageing [10] and in a New York University School of Medicine/Weill Cornell Medical College study [11], and it also limited the decreases in total brain volume over 3 years from age 73 to 76 in a Scottish cohort [12]. When the Mediterranean diet is blended with the “DASH” (Dietary Approaches to Stop Hypertension) diet, termed the MIND (Mediterranean–DASH Intervention for Neurodegenerative Delay) diet, decreased risk of dementia is also observed [13,14]. The MIND differs from the Mediterranean diet by specifying separate categories for green leafy vegetables and berries, with fruit not included, and fish consumption reduced from daily to 2–3 times a week [15]. There is less evidence of the benefits of dietary interventions in patients with existing cognitive decline. However, one pilot study reported a beneficial effect of a coconut-oil-enriched Mediterranean diet on cognitive performance in people with a diagnosis of AD [16]. A decreased risk of dementia has also been associated with adherence to a Japanese diet [17]. Furthermore, adherence to a healthy diet, defined by a modified version of the Alternative Healthy Eating Index, has been associated with a decreased risk of having a ≥3 point decline on the Mini-Mental State Examination Score [18]. The Finnish Geriatric Intervention Study to Prevent Cognitive Impairment and Disability (FINGER) study has also reported that adherence to a healthy diet is associated with better global cognition and executive function [19]. Greater consumption of green leafy vegetables has also been reported to be associated with decreased cognitive decline in the Memory and Aging Project [20], as well as a decreased risk of all-cause mild cognitive impairment (MCI) in the China Longitudinal Aging Study [21]. Diets rich in fruits and vegetable are also associated with better cognitive performance in older adults in a large European study [22], in addition to similar findings in a large study in the United States [23], also confirmed in a meta-analysis [24]. These effects are also observed for subjective cognition [25]. Citrus-rich diets are also associated with decreased dementia risk in a Japanese cohort study [26].

Among specific dietary factors, docosahexaenoic acid (DHA) and omega-3 fatty acids, which are incorporated in the Mediterranean diet, have also been associated with decreased risk of dementia [27]. In the Framingham offspring study, greater red blood cell DHA was associated with decreased risk of both AD dementia and all-cause dementia [28]. Greater DHA is also associated with decreased structural changes on brain imaging in cognitively unimpaired individuals homozygous for the APOE ɛ4 allele [29]. Preservation of memory function and lowered amyloid burden is also observed among APOE ɛ4 carriers [30], and a decreased risk of cognitive decline in multiple domains was also observed with one meal per week of seafood and long-chain n-3 fatty acids in APOE ɛ4 carriers [31]. Greater performance on tasks of fluid intelligence was associated with greater alpha-linolenic acid, stearidonic acid, and eicosatrienoic acid among omega-3 fatty acids in a population of healthy aging individuals [32], and in the Doetinchem Cohort Study, higher alpha-linolenic acid intake was associated with slower cognitive decline in global cognition and memory in middle and older age individuals [33]. However, again, the effects are more limited with dementia already present. A randomized, placebo-controlled trial of high dose omega-3 fatty acids including DHA did not show benefit in patients that had already developed Alzheimer disease [34]. Despite the negative studies in trials of patients with active cognitive decline, though, reviews suggest that supplementation in APOE ɛ4 carriers might be promising for prevention [35].

Several studies have also identified increased lifetime stress as a risk factor for the development of AD. In the animal literature, sustained stress in mice promoted the accumulation of potentially pathogenic levels of phosphorylated tau proteins through glucocorticoid-induced activation of corticotrophin-releasing factor receptors [36]. Studies in the clinical setting, which assessed individuals for the tendency to experience psychological stress and then observed participants for the development of AD, found that individuals who were more prone to psychological stress had twice the likelihood of developing AD [37].

For the association between greater intake of omega-3 fatty acids and lower risk of development of AD, several mechanisms have been proposed. One potential contributor is that omega-3 fatty acids increase SorLA: LR11, a specific sorting protein within the brain, which is found in low levels in AD patients [38]. Furthermore, several studies found that omega-3-fatty-acid-derived oxylipins (e.g., prostaglandins and resolvins) have anti-inflammatory properties which are associated with decreased serum inflammatory cytokines and subsequently protect the brain parenchyma from oxidative stress [39,40]. A more recent study showed that omega-3 fatty acid supplementation in mice fed a diet high in refined carbohydrates resulted in the resolution of elevated serum inflammatory cytokines and prevented the cognitive deficits associated with the carbohydrate diet in non-omega-3-supplemented mice [41].

Therefore, there is considerable evidence to support the notion that stress and low dietary omega-3 fatty acids are risk factors for the development of AD. The aforementioned studies suggest that the anti-inflammatory component of omega-3 fatty acids may be a primary mechanism in preventing AD. In recognizing that stress increases inflammatory mediators and omega-3 fatty acids decrease inflammatory cytokines, further investigation of the interactions between risk factors may be particularly important. This concept has been demonstrated previously by the various components of metabolic syndrome, which combine to increase the risk of cardiovascular disease as well as dementia [42]. Harnessing these principles, models have been developed to predict the risk of developing dementia based upon examination of a broad set of midlife risk factors, including vascular factors [43]. However, most studies have focused on individual risk factors, or a combination of a broad range of factors in a large population, and few have specifically examined the interaction of a specific pair of factors. Due to the importance of the potential interaction between risk factors, our team conducted a pilot study to begin to address this, by exploring whether omega-3 dietary intake and stress exposure interacted to impact the risk of AD.

## 2. Methods

In order to complete this pilot study examining the relationship between stress exposure and dietary omega-3 intake, we recruited participants with Alzheimer’s disease (*n* = 19) and healthy controls (*N* = 23) aged 55–85 that were matched for both age and sex, with half female and half male in both groups. AD participants were recruited from the University of Missouri Memory Disorders Clinic and were included if they met the National Institute of Neurological and Communicative Disorders and Stroke and Alzheimer’s Disease and Related Disorders Association criteria for probable AD, along with a score of 10–24 on the Mini-Mental Status Examination. Controls were subsequently recruited from fliers and senior activities in the community and matched on age and sex. The study was conducted according to the guidelines of the Declaration of Helsinki and approval was sought from the University of Missouri Health Sciences Review Board. Informed consent was obtained from all participants involved in the study.

To evaluate stress, participants were administered the Holmes Rahe Readjustment Rating Scale, which is a 43-item questionnaire assessing stressful life experiences in the last 12 years of the participant’s life, the timeframe monitored in previous studies examining the association between stress exposure and AD [44]. A score of greater than 150 on the Holmes-Rahe Scale suggests a 50% increased risk of having a stress-related illness, while scores of greater than 300 suggest an 80% increased risk [45]. In this survey, participants are given a list of 43 major life stressors and asked regarding the presence or absence of each of these stressors. Each stressor is assigned a score, with death of a spouse receiving a score of 100; divorce, 72; marital separation, 65; death of a close family member, 63; down to minor violation of the law at 11 [45]. While the effect of stress on health was most prominent in the last 6–12 months before the illness in that setting [45], we extended the timeframe of the stress surveys to 12 years, as that was the timeframe for which stress was found to be associated with AD in previous work [37].

As a first-pass effort to examine diet in this setting, in order to evaluate dietary omega-3 intake, participants also completed a 152-item food frequency questionnaire (FFQ) [46] that utilized their reported diet to quantify the estimated daily omega-3 fatty acid content in grams of various foods at a given portion size, then considering the quantity of servings of that portion size ingested by study participants over the past month [47]. The FFQ asks participants about whether they consumed small, medium, or large portions in this semi-quantitative survey. A small serving is ½ of a portion, a medium serving is one portion, and a large serving is 1½ portions. For meat, poultry, and seafood, a medium portion size was listed as 3 oz. Food items were classified into nine categories, including seafood, meats, eggs, milk and milk products, vegetables, fruits, grain products, fats/oils, and legumes. The eight frequency responses were none, once/month, <once/week, 1–2 times/week, 3–4 times/week, 5–6 times/week, daily, and >once/day. The USDA National Nutrient Database for Standard Reference and ESHA Food Processor were used to identify foods containing ≥10 mg omega-3 fatty acid/medium serving. Instructions for using the FFQ included a printed guide showing the fist, palm, hand, and fingers for estimating portion sizes [46]. The FFQ has been shown to be a reliable (alpha coefficient = 0.83) and valid (correlation of omega-3 fatty acid intakes using the food recalls and FFQ = 0.42) measure of omega-3 fatty acid intake in various races and clinical populations [46,47,48,49,50,51].

All surveys for AD patients were completed by a caregiver that has been with the patient over an extended timeframe. To examine whether omega-3 intake interacted with the potential risk of stress, *t*-tests were performed to compare dietary omega-3 fatty acid scores between the AD and control groups in participants that scored greater than 150 (i.e., high stress) as well as those who scored under 150 (i.e., lower stress) on the Holmes Rahe Readjustment Rating Scale.

## 3. Results

In this pilot study, 10 patients with AD and 18 control participants scored greater than 150 on the Holmes-Rahe scale, indicating high levels of life stress. For the high stress analysis, estimated omega-3 fatty acid intake in grams per day was significantly higher in the control group (mean (M) = 1.27 ± 1.21 std dev) among the participants with greater stress exposure than in the AD group (M = 0.615 ± 0.624 std dev) (unpaired Student’s *t*-test *t* = −1.78; *p* = 0.044). The remaining participants scored lower than 150 on the Holmes-Rahe Readjustment Rating Scale, indicating lower levels of life stress. For the low-stress analysis, there were no significant differences in estimated omega-3 fatty acid intake in grams per day between the AD (M = 0.97 ± 0.36 std dev) and control groups (M = 1.37 ± 0.94 std dev) (unpaired Student’s *t*-test *t* = 0.13, *p* = 0.45) for those with scores less than 150 on the Holmes-Rahe Readjustment Rating Scale, indicating less stress exposure (Figure 1). There was no significant correlation between Holmes-Rahe Readjustment Rating Scale scores and estimated omega-3 fatty acid intake across either the AD group (Pearson’s r = −0.21, *p* = 0.4) or the control participants (Pearson’s r = 0.027, *p* = 0.9).

## 4. Discussion

For individuals with significant life stress exposure, estimated daily omega-3 fatty acid intake was higher among individuals not affected by neurodegenerative diseases than among individuals with AD. These findings begin to suggest that greater levels of dietary omega-3 fatty acids may mitigate the effects of increased lifetime stress in the development of AD. The sample size is small, given the pilot nature of the study, but the fact that a difference could be detected with such a modest sample highlights the importance of systematically looking at the interaction between individual AD risk factors in future studies. Larger samples will be needed to build upon these pilot findings for life stress and dietary omega-3 to confirm the findings. As such, these findings should be considered preliminary. Furthermore, the samples are too small to allow for further stratification for other interactions, such as the effects of genetics such as the APOE ɛ4 allele, which appears to be important in the impact of omega-3 fatty acids on dementia outcomes [35]. Additionally, larger samples will be needed to examine the relationships of the findings to AD severity, and interactions with age, educational level, and gender, as well as the effects of medications and how these interact with blood levels of omega-3. It should also be noted that bias can be introduced by community sampling for controls, as individuals interested in participating in research may be more likely to also adhere to healthier diets and lifestyles. Due to the inherent weakness of utilizing dietary surveys, future studies should verify serum omega-3 fatty acid levels and expand on strategies to validate stress exposure history, including the use of surveys that are specifically validated in the AD setting. Since there is no cure for AD at this time, maximizing prevention of AD by minimizing modifiable risks is crucial to decreasing morbidity and mortality related to AD. While modification of one single risk factor may have a modest impact, addressing multiple risk factors may be more effective. This notion is supported by the studies demonstrating the protective effect of the Mediterranean diet, which includes multiple potential protective factors incorporated into one dietary plan, in decreasing the risk of AD.

The literature supports a range of modifiable risk factors for development of AD [1,2,3]. However, what we can state with certainty remains rather limited. This remains an important area that may impact patients and their families. In general, the evidence for potential preventative roles is stronger than for intervention after cognitive impairment already exists. Among specific diets, the evidence is strongest for preventative roles of the Mediterranean diet and MIND diet, and similarly, among specific dietary components, the evidence is strongest for a preventive role for omega-3 fatty acids. Stress exposure and cardiovascular factors are also important. Future work must examine interactions between factors that may have synergistic effects. Also needed is a better understanding of how prevention or interventions may affect cognitive decline generally, as distinguished specifically from that which occurs in particular diseases such as AD, the most common cause of dementia. However, less is known about the interactions between individual risk factors to understand these effects. Larger, systematic studies examining combinations of individual risk factors will be critical for mitigating the risk of AD in the population.

In order to begin to address this, our pilot study suggests that dietary omega-3 intake and stress exposure may interact in the risk of developing AD. Despite the limitations of this study, it provides a first step in the direction of a systematic understanding of the interactions between individual modifiable factors in the development of AD.

## Figures and Tables

**Figure 1 biomolecules-13-01096-f001:**
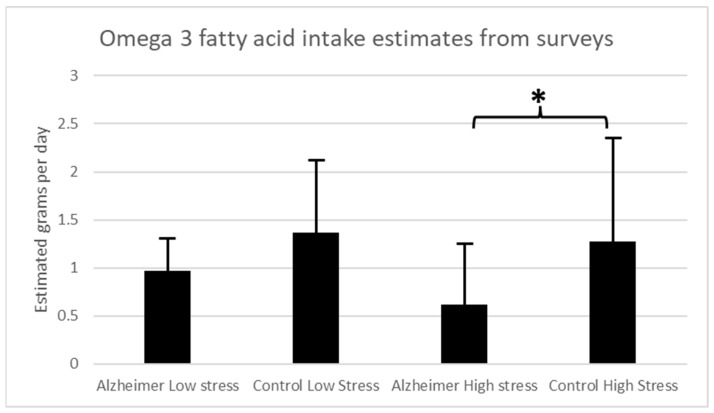
Estimated omega-3 fatty acid levels in grams per day based on dietary surveys for AD patients and controls for those with high stress exposure (>150 on stress survey) and low stress exposure (<150 on stress survey). (* *p* < 0.05).

## Data Availability

Data will be made available up on request.

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
