# Peer review of "Potential Neuroprotective Effects of Dietary Omega-3 Fatty Acids on Stress in Alzheimer’s Disease"

_biomolecules, 2023, doi:10.3390/biom13071096_

Round 1

Reviewer 1 Report

The aim of this pilot study was to evaluate the potential neuroprotective effects of omega3 fatty acids on Alzheimer patients.

The topic is very interesting, potentially suggesting a modifiable factor in the development of Alzheimer disease. Indeed, the results suggest that greater levels of dietary omega 3 fatty acids may mitigate the effects of increased stress in the development of Alzheimer disease.

There are some limitations in this work, as correctly highlighted by the authors: 1) the sample size is small; 2) the evaluation of omega-3 fatty acids should be verified in serum, and not only by FFQ questionnaires.

However, because this is a pilot study, and having highlighted important results, even in the presence of the limitations described above, I think that the work can be published.

In the future studies, with a larger sample size, I would recommend the authors to evaluate any gender effect:  a gender analysis will definitely improve the work.

Author Response

Thank you for your helpful comments and recognition of the importance of the question at hand!  We have added further comment on the need for a larger sample size and the very important need to explore gender effects with this larger sample.  Our current sample is too small for further meaningful subtyping.

Reviewer 2 Report

In this manuscript the authors present an epidemiologic study analyzing the interaction between two parameters -stress and omega-3 fatty acids in diet- in association with Alzheimer’s disease (AD). The results suggest an association between low omega-3 intake and AD in high stress exposure.

The study is interesting and may pave the road to further and more comprehensive studies on the interactions between risk factors for AD. It presents several limitations. As stated by the authors, this is a pilot study, with conclusions being preliminary. Below are my comments and suggestions:

-a mote precise demographic information about patients should be provided, like age ranges, median or average age per group (controls, AD, high stress, low stress), number of males and females, etc.

-is there any association between omega-3 intake and age? Between omega-3 intake and gender? Are there any other available demographic parameters that could be considered?

-Is it possible to consider different levels of AD severity or progression and their association with stress/omega-3 intake?

-the HRR rating scale should be further explained, if possible include the questionnaire (as well as the FFQ questionnaire) as supplemental material.

-A correlation test could be established between the HRR score and the omega-3 intake.

-“grams per day” should be added as title of the y axis in the figure

-I do not think that “in aging” in the title is relevant, as aging is not addressed by the study

Author Response

We thank the reviewer for recognizing the importance of this question.  To our TREMENDOUS frustration, or demographics file was corrupted, limiting our ability to report demographics, but we did individually match by age, and were able to identify gender, so we do report that.  Therefore, we could not do analysis of age and omega-3 intake, but do discuss this as critical future directions in the modified paper, along with severity effects.  Medication effects should also be tracked in future studies, as well as educational level, as we now comment.  We agree it is very helpful to provided expanded description of the HRR and the FFQ for readers, and we did this.  Copyright limits our ability to provide the actual tables, though.  We did examine and report the HRR and omega-3 correlation- this is an excellent point.  We also changed the Figure axis label as suggested, which helps clarity, and modified the title as suggested.

Reviewer 3 Report

In this study, the authors performed a pilot study surveying patients with AD and controls for stress exposure and dietary omega-3 fatty acid intake to explore their relationship for risk of AD. For individuals with the greatest stress exposure, omega-3 fatty acid intake was significantly greater in healthy controls than in AD patients. There was no difference among those with low stress exposure. These pilot results suggest that omega-3 fatty acids may mitigate AD risk in the setting of greater stress exposure. The data of this study are interesting although the sample size is too small. I have some minor comments.

1) Please add Table 1 of clinical variables and medication information of participants in this study.

2) Please add the limitation (i.e., sample size, medication, blood levels of omega-3) of this study.

Author Response

We thank the reviewer for recognizing the importance of this question.  To our TREMENDOUS frustration, or demographics file was corrupted, limiting our ability to report demographics, but we did individually match by age, and were able to identify gender, so we do report that.  We also discuss the limitation and that medication effects should also be tracked in future studies with larger samples, as well as educational level, as we now comment, with the need for how all of these factors relate to blood levels of omega-3.